# Effects of Protein Supplementation During High-Intensity Functional Training on Physical Performance in Recreationally Trained Males and Females: A Randomized Controlled Trial

**DOI:** 10.3390/nu17091441

**Published:** 2025-04-25

**Authors:** Christina Karpouzi, Ioannis Kosmidis, Anatoli Petridou, Gabriela Voulgaridou, Sousana K. Papadopoulou, Gregory C. Bogdanis, Vassilis Mougios

**Affiliations:** 1Laboratory of Evaluation of Human Biological Performance, School of Physical Education and Sport Science at Thessaloniki, Aristotle University of Thessaloniki, 54124 Thessaloniki, Greece; kosmidia@phed.auth.gr (I.K.); apet@phed.auth.gr (A.P.); mougios@auth.gr (V.M.); 2Department of Nutritional Sciences and Dietetics, International Hellenic University, 57400 Thermi, Greece; gabivoulg@gmail.com (G.V.); sousana@the.ihu.gr (S.K.P.); 3School of Physical Education and Sport Science, National and Kapodistrian University of Athens, 17237 Athens, Greece; gbogdanis@phed.uoa.gr

**Keywords:** protein intake, nutrition, high-intensity functional training, performance, strength

## Abstract

**Background/Objectives**: High-intensity functional training (HIFT) combines multijoint aerobic and resistance exercises. Despite its popularity, limited research has investigated dietary or supplementation strategies to enhance adaptations to HIFT. Hence, this study aimed to examine the effects of egg white and whey protein supplementation during HIFT on physical performance in trained individuals. **Methods**: Thirty recreationally trained volunteers (20 males, 10 females), aged 23–55, underwent 6 weeks of HIFT (three times/week) while receiving 0.6 g/kg/day of egg white protein, whey protein, or maltodextrin (placebo) in a researcher-blinded, randomized, triple-crossover, and counterbalanced design, with 2 weeks of washout between supplements. Participants followed isoenergetic diets providing 1.0 g/kg/day of protein. Before and after each intervention, VO_2_max, the maximal strength (1 RM) and force–velocity relationship of shoulder press, the peak torque and strength endurance of knee extensors and flexors, and the strength endurance of core muscles were measured. The training session load was monitored during each intervention period’s first and last weeks. Data were analyzed by three-way ANOVA (supplement × time × sex), with repeated measures on supplement and time. **Results**: The duration, energy expenditure, training load score, and cardio load of each training session increased from the beginning to the end of each training period by 2–11% (*p* < 0.05). The 1 RM of shoulder press and strength endurance of core muscles increased by 3–6% (*p* < 0.001). Protein supplementation did not affect any of these outcomes. **Conclusions**: Short-term HIFT improved exercise capacity, upper-body strength, and core endurance. However, increasing protein intake from 1.0 to 1.6 g/kg/day did not further enhance performance.

## 1. Introduction

Physical performance encompasses multiple parameters including muscular strength, power, and endurance; aerobic capacity; speed; and agility. These attributes can be efficiently developed by a variety of training methods, including resistance training and high-intensity interval training [1]. A newer version of the latter, named high-intensity functional training (HIFT), incorporates the principles of high-intensity interval training alongside the template of CrossFit^®^ [2]. HIFT is characterized by multimodal movements that include bodyweight exercises and weightlifting derivatives, mimic real-life activities, are performed at high intensity, and capitalize on the progression principle [1,2,3]. HIFT seems to have the potential to elicit positive and versatile adaptations in the musculature and metabolism in a time-efficient manner [1,4,5,6,7].

As HIFT gains recognition and popularity, participants are increasingly interested in dietary practices and supplements to enhance adaptations. A recent survey of 2576 CrossFit practitioners reported that 82% of the responders consumed at least one dietary supplement, with protein topping the list (51%) [8]. Similarly, a survey of 112 CrossFit practitioners found the prevalence of supplement use to be 78%, with whey protein being the most common supplement (used by 63% of the responders) [9]. However, we could find no scientific documentation of the benefits of protein supplementation in HIFT, in contrast to its well-documented effectiveness in enhancing muscle hypertrophy and strength when combined with resistance training [10,11]. Moreover, to our knowledge, no study has examined sex-related differences in the responses to HIFT, not to mention protein supplementation during HIFT.

Whey is by far the most widely used source of protein supplements in sport [9,10] due to its low cost and high quality. An alternative source is egg white, which is marketed in various products as a substitute for whole eggs in cooking and confectionery or as a convenient, high-quality protein supplement for athletes. Research on the utility of egg white supplementation in training is scant. Bagheri et al. [12,13] studied egg white supplementation during 12 weeks of resistance training, albeit without a non-protein control (the comparator was whole eggs of the same protein content as egg whites), while Hida et al. [14] found no differences between egg white and maltodextrin supplementation in the body composition or muscle strength of athletes after 8 weeks of regular training.

Considering the dearth of research on protein supplementation in HIFT and on egg white supplementation in training generally, the present study aimed to compare the effects of egg white, whey protein, and maltodextrin (as placebo) supplementation on THE physical performance parameters of recreationally trained males and females undergoing HIFT. We hypothesized that the two protein supplements would enhance the effects of training on physical performance and that the two sexes would respond differently to training and/or supplementation.

## 2. Materials and Methods

### 2.1. Study Design and Ethics

A randomized, triple-crossover, and counterbalanced design was employed to examine the effects of protein supplementation on physical performance parameters of recreationally trained males and females who underwent 6 weeks of HIFT, with 2 weeks of washout between supplements. The allocation sequence was generated through the drawing of lots by one of the authors, who also assigned the participants to the intervention. The duration of HIFT was based on the existing literature, which considered 6 weeks to be the minimal duration of a resistance exercise program to produce measurable adaptations [10,11] and 4 weeks to be the corresponding minimal duration of HΙFΤ [7]. Thus, the duration of the study (excluding the initial and final measurements) was 22 weeks, starting on 29 November 2021 and ending on 1 May 2022.

The study was blind regarding the intervention providers (HIFT instructors and supplement distributors), outcome assessors, and investigators; it was not blind regarding the participants due to the different forms of egg white (liquid) from that of whey protein and maltodextrin (powder), although they could not discriminate the latter two. The participants were asked not to disclose which supplement they were receiving at any time to the intervention providers, outcome assessors, and investigators.

Participants underwent measurements before the first intervention period, during the two washout periods, and after the third intervention period (Figure 1). The study design was approved by the Ethics Committee of the Aristotle University of Thessaloniki (approval number 14435/19-1-2021), was registered in ClinicalTrials.gov (NCT05396963/26-4-2022), and was carried out according to the Declaration of Helsinki.

### 2.2. Participants

The sample size was calculated a priori using the G*Power software (version 3.1.9.2, Kiel University, Kiel, Germany). To detect significant effects with a medium effect size of 0.059 (as η^2^ for factorial ANOVA) [15], α of 0.05, power of 0.8, correlation coefficient between repeated measures of 0.5, and two groups of participants (males and females), a sample size of 20 was required. To account for possible dropouts (especially due to the COVID-19 pandemic), we decided to recruit 40 participants.

Participants were recruited by the first author voluntarily among gym clients and through social networks and personal acquaintances. Interested parties were asked to complete an online questionnaire; those who met all inclusion criteria and did not meet any exclusion criterion were informed about the study through a detailed text. The inclusion criteria were as follows: age 18–65, regular training (mixed endurance and resistance training 3–6 times a week, 50 min each session, for at least the past 4 months), clearance from a pathologist or cardiologist to perform maximal exercise, and mixed isoenergetic diet for the past 4 months. The exclusion criteria were smoking, musculoskeletal injury or chronic disease, egg and milk allergy, pregnancy, lactation, regular use of prescription medicine or supplements that might affect muscle function or recovery over the past month, intermittent or religious fasting, and any ketogenic or protein diet.

Based on the aforementioned criteria, we were able to recruit 22 males and 18 females. To ensure the independence of observations, we did not include participants in a group who were relatives or lived in the same household.

During their first visit to the laboratory, the participants received detailed written and oral information about the study’s purpose, design, measures, and possible risks. Then they provided written consent to participate, accompanied by an agreement to adhere to the instructions and give notification of any noncompliance. Subsequently, we measured their body weight to the nearest 0.1 kg with minimal clothing using an electronic balance (Seca, Hamburg, Germany) and height to the nearest 0.5 cm with a stadiometer fitted to the balance.

### 2.3. Supplementation

During the three 6-week intervention periods, the participants received daily either an egg white supplement (E, EGGPRO Muscle, Avgodiatrofiki, Nea Santa, Greece) in quantities providing 0.6 g protein per kg body weight, a whey protein supplement (W, Warriorlab Complete Whey, Warriorlab, Athens, Greece) in quantities providing 0.6 g protein/kg, or a maltodextrin supplement (M, Warriorlab Maltodextrin, Warriorlab) in quantities providing 0.6 g maltodextrin/kg (equivalent to the carbohydrate content of a large banana). According to data obtained from the US Department of Agriculture FoodData Central databases for egg white and from the manufacturer for whey protein, the two supplements contained, as percentages of the total amino acids, 8.1 and 10.7% leucine, respectively. Their total essential amino acid contents were 43.3 and 47.0%.

The supplements were taken in two equal daily doses, approximately 12 h apart. E was provided as a strawberry-flavored, pasteurized, liquid product, commercially available and consumed as is. W and M were provided in powder form and were dissolved in water before ingestion. The timing of supplementation relative to the training sessions was not controlled, since it has been shown not to play a significant role in the long-term adaptations to resistance training [16], and no such information is available regarding HIFT. The participants were asked to record each dose taken on specially prepared logs. They did not receive any supplements during the washout periods.

### 2.4. Nutritional Control

Participants received individualized isoenergetic dietary plans throughout the entire study period, providing 1.0 g protein/kg/d (thus reaching a total of 1.6 g/kg/d when they consumed E or W), about 55% of energy from carbohydrate, and about 35% of energy from fat. The implementation of the dietary plans began one week before the onset of supplementation and stopped one week after the end of supplementation. The dietary plans were changed every 6 weeks for variety. Every week, we asked each participant what they thought their percentage compliance with the dietary plan was.

### 2.5. Monitoring of Menstrual Status

Throughout the study, female participants kept a menstruation diary. This information was used to calculate the day of the cycle when each assessment was performed.

### 2.6. Exercise Protocol

The exercise protocol was carried out in two well-equipped commercial gyms offering identical group programs. Participants were asked to join any of the available HIFT programs at a frequency of three times per week, with the option to attend different programs from time to time. This freedom was given to increase the ecological relevance of the study as it reflects a usual practice of exercisers attending gym programs. The HIFT programs included multimodal patterns of movement, combining both endurance and strengthening exercises with equipment such as TRX, BOSU, kettlebells, and barbells.

Certified instructors and/or research assistants, all carefully briefed by the investigators, supervised all sessions and checked the proper application of the exercise protocol, including the “work to momentary failure” principle. The participants did not perform any intense training (that is, above 76% of HRmax) during the remaining four days of the week. The duration of the workout was about 45 min, including 5 min of warm-up and 5 min of cool-down, and could be modified according to the progression of the participants’ performance. During the washout periods, the participants continued their usual training (what they did before joining the study).

### 2.7. Outcome Measures

Training variables were assessed at the gyms at the beginning and end of each intervention period. Performance measures were assessed at the Laboratory of Evaluation of Human Biological Performance at the beginning of the study (within the two weeks preceding it), during the washout periods, and within the week following the end of the intervention. Each time, the performance measurements were separated into two days to ensure that the participants could perform the tests to their full potential and were performed at least three days after the last training session of each intervention period to avoid any acute effects on the outcome measures. The order of the measurements was as follows: On the first day we assessed grip strength, the force–velocity relationship of knee extensors and flexors, the maximal dynamic strength of shoulder muscles, and aerobic capacity. On the second day we assessed the strength endurance of knee extensors and flexors, the force–velocity relationship of shoulder muscles, and the strength endurance of core muscles. At the beginning of each day, the participants executed a general warm-up that included 5 min on a cycle ergometer at a moderate intensity and dynamic stretches for the muscle groups that were going to be assessed.

The following outcome measures were assessed.

### 2.8. Training Variables During HIFT Sessions

During an exercise session of the first week and an exercise session of the final week of each intervention period, each participant’s heart rate (HR) was monitored using telemetric sensors and software (Polar Team Pro GPS telemetric system, v. 2, Polar, Kempele, Finland). The following variables were obtained from the software’s output: total exercise time, average HR, highest HR, exercise time in each of 5 HR zones based on percentages of HRmax (zone 1, 50.1–60; zone 2, 60.1–70; zone 3, 70.1–80; zone 4, 80.1–90; zone 5, 90.1–100), percentage of total exercise time expended in each of those HR zones, total energy expenditure, training load score (an index of the strenuousness of the training session, calculated based on its intensity and duration), and cardio load (calculated from the HR data and duration of the session) [17].

### 2.9. Maximal Dynamic Strength of Shoulder Muscles

The maximal dynamic strength of shoulder muscles was assessed by determining one-repetition maximum (1 RM), according to the procedures outlined by the US National Strength and Conditioning Association [18]. The participants received standardized instructions on the proper technique for performing the shoulder press on a Smith machine, with the bench positioned near-vertically to support the head and torso. Feet rested on the floor, and the knee angle was approximately 90°. After a general warm-up of the shoulder muscles and stretching exercises, the subjects were instructed to perform 10 repetitions at 50% of the estimated 1 RM, 4–6 repetitions at 75%, and 2–4 repetitions at 85%. Then, the resistance was gradually increased by 2.5–10 kg, and subjects performed the shoulder press until they could perform only one repetition. The recovery period was 2 min between the first two sets and 4 min between subsequent sets. Participants were verbally encouraged to do their best, and the investigator was present to ensure the safe and correct execution of the procedure.

### 2.10. Force–Velocity Relationship of Shoulder Muscles

The force–velocity relationship of the shoulder muscles was assessed by measuring the peak velocity and peak force of shoulder press in a Smith machine with a linear position transducer (Tendo Power Analyzer System v. 314, Trencin, Slovak Republic). Participants lifted weights corresponding, as closely as possible, to 30, 45, 60, and 75% of 1 RM for males and 45, 55, 65, and 75% of 1 RM for females. Participants performed 3 repetitions at each weight as fast as possible, with 3 min of rest between sets. The average peak velocity and average peak force of the 3 repetitions with each weight were plotted to form a force–velocity graph. From this, the y-intercept (corresponding to the force at zero velocity and considered the maximal force) and the x-intercept (corresponding to the velocity at zero force and considered the maximal velocity) were calculated.

### 2.11. Grip Strength

The grip strength was measured in the dominant hand using a hand dynamometer (Takei, Osaka, Japan). Participants stood with the arm by their side and the elbow flexed at 90° with neutral wrist position. They were instructed to perform 3 maximal efforts of 5 s under verbal encouragement, with 1 min of rest in between. The best effort was recorded.

### 2.12. Strength Endurance of Core Muscles

Strength endurance of core muscles was assessed by the number of continuous sit-ups performed within 60 s. The participants started from the supine position, with arms crossed in front of the chest, knees bent, and feet secured by the investigator, who verbally encouraged the procedure and counted the repetitions. To complete a correct repetition, the participants had to lift the torso and reach the knees with the elbows.

### 2.13. Force–Velocity Relationship of Knee Extensors and Flexors

The force–velocity relationship of knee extensors (quadriceps) and flexors (hamstrings) was assessed through isokinetic dynamometry (HUMAC NORM 770, Stoughton, MA, USA), using maximal concentric contractions at angular velocities of 60, 120, 180, and 240°/s. Participants were seated and secured to the dynamometer with chest and thigh straps, with their torso upright against the vertical back of the seat. The dynamometer’s axis was adjusted so that the center of motion of the lever arm was aligned with the flexion–extension axis of the knee joint. After a general warm-up on a cycle ergometer (at 50–60 W for about 5 min) and dynamic stretching exercises, participants performed 3 maximal concentric extensions and flexions at each angular velocity, with 1 min of rest between angular velocities, under strong verbal encouragement. The peak torque was determined as the highest value of knee extension and knee flexion obtained at each angular velocity.

### 2.14. Strength Endurance of Knee Extensors and Flexors

The strength endurance of the knee extensors and flexors was assessed through a fatigue protocol in the isokinetic dynamometer, consisting of 30 repeated maximal concentric extensions and flexions at 180°/s under strong verbal encouragement. Body positioning was as described above. Strength endurance was defined as the ratio of the total work of the last 15 efforts to the total work of the first 15 efforts, multiplied by 100 (endurance ratio or fatigue ratio).

### 2.15. Aerobic Capacity

Aerobic capacity was assessed by measuring the maximal oxygen uptake (VO_2_max) through a maximal graded exercise test on a horizontal treadmill, connected with an ergospirometer (Jaeger Oxycon Pro, Würzburg, Germany). The initial treadmill speed for males was 6 km/h and was increased every minute by 1 km/h until exhaustion. Females performed the same protocol, except that the initial speed was 5 km/h. Oxygen uptake was measured breath by breath and was averaged every five breaths. HR was monitored using a Polar HR monitor (Polar, Kempele, Finland). The test completion criteria were any two of the following: a plateau in oxygen uptake despite increasing speed (<150 mL/min or <2.1 mL/kg/min with an increased stage), a respiratory exchange ratio above 1.10, and an HR within 10 beats/min of the age-predicted maximal (HRmax = 220 − age).

### 2.16. Adverse Events

Some participants missed measurements due to adverse events, personal circumstances (workload or relocation), failure to attend measurement sessions, technical issues with the measuring equipment, or pregnancy. The analysis of adverse events revealed that 12 participants contracted COVID-19 (apparently unrelated to supplement use or training), two experienced gastrointestinal disturbances (while taking W and P), and one was injured in a sports activity unrelated to the study. These adverse events are unlikely to have influenced the validity of the comparison of the supplements since they were evenly distributed between supplements.

### 2.17. Handling of Missing Data

Of the 40 participants, 20 had complete datasets, 9 missed one or two measurements, and 11 dropped out between weeks 2 and 21. We aimed to perform an intention-to-treat (ITT) analysis, which included data from all enrolled participants to minimize the bias introduced by analyzing only those who adhered to and completed the treatment originally allocated (per protocol analysis) [19,20]. However, ITT presents challenges when dealing with missing data due to dropouts, no-shows, or technical problems. In ITT analysis, the missing outcomes are estimated from available data using various methods [21]. In the present study, 10 participants withdrew between weeks 2 and 14, resulting in the data obtained from each one being fewer than or equal to the missing data. Due to the complex design of the study (administering 3 different supplements during different periods), we considered it too arbitrary to fill in the missing data of these participants by any method. Hence, we excluded these 10 participants from the analysis. For one participant who withdrew from the study at week 21 (just before the final measurements) and for the 9 participants with few missing data, we applied imputation methods as follows: missing data at the beginning of the study were replaced with the value from the next measurement, missing data in the middle of the study were replaced with the mean of the previous and next values, and missing data at the end of the study were replaced with the value from the previous measurement (the so-called last-observation-carried-forward method). The causes of the missing data were not related to the research questions of the study. Therefore, these data could be characterized as “missing completely at random” [21], and using the obtained or inferred data for analysis was expected to yield unbiased conclusions.

In essence, we adopted a mixed model of data handling, representing a reasonable compromise between the ITT and PP approaches. Consequently, the analysis included 30 participants: 20 males and 10 females.

### 2.18. Statistical Analysis

Outcome measures are presented as the mean ± standard deviation (SD). Differences in the number of training sessions during intake of each supplement, doses of each supplement taken, and adherence to the dietary plans during intake of each supplement were examined by 2-way analysis of variance (ANOVA) with three levels of supplementation (E, W, P), two levels of sex (male and female), and repeated measures on supplementation. Differences in the days of the menstrual cycle on which measurements were performed pre- and post-supplementation were examined by 2-way ANOVA with three levels of supplementation, two levels of time (before and after each intervention period), and repeated measures on both factors. To analyze the effects of supplementation, time, and sex on the outcome measures, we performed 3-way ANOVA with three levels of supplementation, two levels of time, two levels of sex, and repeated measures on supplementation and time. To explore the presence of any order effect on the outcome measures, we performed the same 3-way ANOVA by replacing supplementation with the order of supplementation (1st, 2nd, and 3rd period). Sphericity was tested with Mauchly’s test, and when violated, the Greenhouse–Geisser correction was applied. The effect sizes (ESs) for the main effects and interactions were determined as partial η^2^ and were classified as small (0.01–0.058), medium (0.059–0.137), or large (>0.137) according to Cohen [15]. Correlations between variables were examined with Pearson’s correlation analysis. Statistical significance was declared at α = 0.05. Statistical analyses were performed in SPSS, version 28.0 (IBM SPSS Statistics, Armonk, NY, USA).

## 3. Results

### 3.1. Baseline Characteristics

Table 1 presents the baseline characteristics of the participants.

### 3.2. Examination of Confounding Variables

We examined three potential confounding variables: the number of training sessions during the intake of each supplement, the doses of each supplement taken, and adherence to the dietary plan during the intake of each supplement. No statistically significant outcome was found. Adherence to training sessions was 91 ± 14%, adherence to supplementation was 98 ± 5%, and compliance with dietary plans was 69 ± 19%, all independent of supplement and sex.

### 3.3. Menstrual Status

From the menstruation diaries completed by 9 of the 10 female participants (one did not provide such information), we found no significant difference between E, W, and P supplementation; no significant difference between the pre- and post-intervention evaluations; and no significant interaction of supplement and time regarding the day of the cycle on which the participants were evaluated (*p* > 0.05).

### 3.4. Training Variables of HIFT Sessions

Table 2 presents the descriptive statistics of the training variables during two HIFT sessions, one on the first week and one on the final week of each intervention period. Table 3 presents the inferential statistics of the variables that produced significant main effects and/or interactions. The results involve 22 participants (13 males and 9 females). For practical reasons, it was not possible to evaluate the remaining eight participants at any session.

The results showed a significant increase in the duration of the training session from 41:40 ± 06:43 min:s at the start to 46:25 ± 08:40 min:s at the end of each intervention period (by 11%), independent of supplementation or sex, with a large ES. The same happened with the time in zone 2 (60.1–70.0% HRmax), which increased from 07:53 ± 06:43 min:s to 09:03 ± 05:39 min:s, with a large ES. Also, the time in zone 4 (80.1–90% HRmax) increased significantly from 10:51 ± 05:33 min:s to 12:26 ± 06:11 min:s, with a large ES.

Energy expenditure showed significant main effects of time and sex, as well as an interaction of the two factors. The significant interaction could be explained by the fact that although both males and females increased their energy expenditure, the increase in males (from 488 ± 144 to 563 ± 130 kcal, by 15%) was higher than that in females (from 359 ± 88 to 368 ± 109 kcal, by 2%).

The training load score and cardio load increased from the start to the end of each intervention period independent of supplementation or sex, with a large ES. The training load score increased from 77 ± 28 to 84 ± 26 (by 9%) and the cardio load from 70 ± 26 to 76 ± 24 (also by 9%).

### 3.5. Performance Variables

Table 4 presents the descriptive statistics of the performance variables assessed. Table 5 presents the inferential statistics of the variables that produced significant main effects and/or interactions. The results showed a significant increase in 1 RM of shoulder press from 55.4 ± 20.1 kg at the beginning of each intervention period to 56.9 ± 20.3 kg at the end of each intervention period (by 3%), regardless of supplement or sex, with a large ES. There was a triple interaction in the maximal velocity of shoulder muscles, which could be explained by an increase in males and a decrease in females while taking E, an increase in males and practically no change in females while taking W, and a decrease in males and increase in females while taking P. There was also a significant increase in the strength endurance of core muscles from 39 ± 6 sit-ups/min at the beginning of each intervention period to 41 ± 6 sit-ups/min at the end of each intervention period (by 6%), with a large ES.

We found a small (4%) but significant decrease in the peak torque of the knee extensors at 60 °/s, from 179 ± 58 Nm at the beginning of each intervention period to 172 ± 52 Nm at the end of each intervention period, with a large ES. There was also a time-by-sex interaction with a large ES, which can be explained by the fact that the reduction in males (from 209 ± 46 to 200 ± 40 Nm) was larger than the reduction in females (from 118 ± 22 to 116 ± 18 Nm). We also found a small (2%) but significant decrease in the peak torque of knee extensors at 120°/s, from 157 ± 48 Nm at the beginning of each intervention period to 153 ± 47 Nm at the end of each intervention period, with a medium ES. Regarding the endurance ratio of knee extensors, there was an interaction of time and sex with a large ES, which can be explained by the fact that in males, there was an increase from 73 ± 9% at the beginning of each intervention period to 74 ± 10% at end of each intervention period, while in females, there was a decrease from 72 ± 8% to 70 ± 8%. Finally, significant differences between sexes were found in most of the performance parameters, with males having higher values than females.

### 3.6. Order Effect

We found a significant order effect on the following outcome measures:The duration of each training session (in min:s) was 42:46 ± 6:43 in the first intervention period, 41:58 ± 5:56 in the second period, and 47:24 ± 10:03 in the third period (*p* = 0.002, ES = 0.315).The average HR in each training session (in bpm) decreased from 141 ± 11 in the first period to 140 ± 13 in the second period to 135 ± 14 in the third period (*p* = 0.026, ES = 0.166).The same pattern was depicted in the average HR as a percentage of HRmax, which decreased from 76 ± 6 in the first period to 75 ± 8 in the second period to 73 ± 8 in the third period (*p* = 0.040, ES = 0.149).The time expended in HR zone 2 during each training session (in min:s) increased from 07:25 ± 04:38 in the first period to 07:45 ± 4:24 in the second period to 10:13 ± 6:00 in the third period (*p* = 0.028, ES = 0.192).The time expended in HR zone 3 (in min:s) was 12:13 ± 04:03 in the first period, 10:02 ± 04:26 in the second period, and 12:28 ± 6:43 in the third period (*p* = 0.040, ES = 0.149).The time expended in HR zone 4, as a percentage of total exercise time, decreased from 28 ± 12 in the first period and 28 ± 11 in the second period to 22 ± 13 in the third period (*p* = 0.031, ES = 0.212).1 RM of shoulder press (in kg) increased from 54.4 ± 19.8 kg the first period to 56.8 ± 20.3 in the second period to 57.4 ± 20.6 in the third period (*p* < 0.001, ES = 0.371).The strength endurance of core muscles (in sit-ups/min) increased from 38 ± 6 in the first period to 40 ± 6 in the second period to 42 ± 6 in the third period (*p* < 0.001, ES = 0.638).

### 3.7. Correlations Between Study Variables

We only found a few scattered significant correlations between different study variables, most likely due to chance. On the other hand, the repeated measures of all performance variables (that is, at the beginning and end of each intervention period), which had been assumed to have a correlation coefficient of 0.5 for the a priori calculation of sample size (see above under the Materials and Methods, Participants Section), had an average correlation coefficient of 0.82, that is, well above the assumed value. Based on this value, the study was sufficiently powered to detect significant effects with an effect size (η^2^) as low as 0.021 (small).

## 4. Discussion

The present study investigated the effects of protein supplementation during short-term HIFT on the physical performance parameters of recreationally trained males and females. One main finding was that 18 sessions over 6 weeks resulted in increased exercise capacity during training, maximal upper-body strength, and strength endurance of core muscles. However, protein supplementation (in the form of either egg white or whey protein) that raised the daily protein intake from 1.0 to 1.6 g/kg did not confer any additional benefit.

The monitoring of the training variables during a session in the first and the last weeks of each intervention period revealed a large increase in the total exercise time. This was mainly due to the participants’ ability to perform more repetitions to exhaustion, indicating enhanced exercise capacity. Yet, the average HR did not change significantly (or even numerically, averaging 139 bpm in both the first and last sessions). This suggested a positive adaptation of the cardiovascular system to short-term HIFT. Thus, the observed increases in energy expenditure, training load score, and cardio load should be attributed to the longer duration of training sessions. The only other study [22] that examined the effect of HIFT on HR variables during training sessions showed (similarly to our study) no change in the highest HR or time in zone 5 but (contrary to our study) a decrease in the average HR. This difference may be due to the longer duration (8 weeks) and higher intensity (resulting in an average HR of 158 bpm) of HIFT in that study.

The absence of a training effect on VO_2_max in the present study is in accordance with the results of Sobrero et al. [23] (who reported no change in VO_2_max after 6 weeks of HIFT or traditional circuit training in recreationally active females) and those of Outlaw et al. [24] (who reported no change in VO_2_max in CrossFit participants). However, other studies investigating the impact of HIFT on aerobic capacity have demonstrated increases in VO2max in both trained and inactive individuals [25,26,27,28,29]. These discrepancies could be attributed to factors such as the participants’ training status, the relatively short duration of the intervention, the specific training protocols employed, and sex, although none of these factors alone distinguishes the two groups of studies.

Our findings of an improvement in upper-body strength and core endurance with HIFT can be linked to the progressive increase in the duration of training sessions as participants gradually performed more repetitions of each exercise, often with greater resistance. These results agree with the findings of several studies with varying duration of HIFT and participant characteristics [22,26,28,29,30,31,32].

The absence of improvement in lower-body strength, evidenced by no significant changes in eight isokinetic variables and a decrease in two (peak torque of knee extensors 60 and 120°/s; Table 3 and Table 4), may be attributed to the nature of HIFT, which predominantly involves multijoint exercises. Such exercises may have a limited impact on performance in monoarticular testing. This may be a limitation of the study, highlighting the need for further research incorporating more functional testing methods. Another consideration may be the change in dietary plans every six weeks during the study, implemented for variety and to reflect real-world variability (especially considering seasonal changes in food availability). This is a regular practice followed by dietitians to prevent boredom and increase compliance. Nevertheless, the dietary plans differed only in food sources, not macronutrient distribution (which was the same), and were always isoenergetic. This choice of design reflects a balance between applicability, ecological validity, and experimental control.

While numerous studies have extensively examined the role of protein supplementation in enhancing adaptations to resistance training [10,11,33] or endurance training [34], there remains a notable gap in the literature regarding the effects of protein supplementation in the context of HIFT. This training modality, characterized by multimodal movements performed at high intensities, imposes unique physiological demands that may interact differently with protein supplementation. To our knowledge, this is the first study in which HIFT was combined with protein supplementation. The inclusion of a relatively understudied protein source (egg white) in a triple-crossover design, comparing it with whey protein and maltodextrin, enhances the impact of our study.

Despite the adaptations in physical performance elicited by HIFT, protein supplementation did not provide any additional benefits. Given the recommended daily protein intake of at least 1.6 g/kg for resistance training [10,11], it was reasonable to anticipate that increasing the daily protein intake from 1.0 to 1.6 g/kg would enhance the effects of HIFT on the study’s outcome measures. Nevertheless, this did not prove to be the case, thus rejecting our hypothesis, as stated in the introduction. Thus, our results indicated that a daily protein intake of around 1.0 g/kg/d was adequate to support adaptations to HIFT. The absence of an effect of protein supplementation on performance agrees with the findings of a recent study from our research team that involved 10 weeks of Pilates training in females. Similarly to the present study, although the training intervention improved performance (core muscle endurance and joint flexibility), supplementation with 0.6 g of whey protein per kilogram body weight daily had no additional effect compared with placebo [35].

To our knowledge, this is the first study that compared males and females in the context of HIFT. The differences in energy expenditure during training sessions (Table 2) and most performance parameters (Table 4) could be attributed to established biological differences between sexes. The few different responses of the two sexes to HIFT, based on the significant interactions, were mixed and did not point to a uniformly higher response of one sex against the other.

Last, the analysis of the order effect highlights the positive adaptations to three 6-week periods of HIFT, in terms of the duration of training sessions, HR, 1 RM of shoulder muscles, and strength endurance of core muscles. This confirms the importance of and justifies our choice to employ a counterbalanced design. Furthermore, the fact that these additive adaptations to the three HIFT periods were practically the same as those to the single HIFT periods, as summarized at the beginning of the discussion, demonstrated that six weeks of training were sufficient, at least as far as the studied performance parameters were concerned.

## 5. Conclusions

The present study demonstrated that short-term HIFT improved aspects of physical performance in healthy, recreationally trained males and females. From a practical perspective, our findings are added to the existing body of knowledge showing that HIFT can be regarded as an effective training method for promoting fitness adaptations. Moreover, our approach of allowing a variety in training contents within the limits of HIFT strengthens the generalizability and applicability of the findings to real-life settings. Importantly, we showed for the first time that protein supplementation did not influence these adaptations, suggesting that a modest daily protein intake in the context of a balanced and isoenergetic diet was sufficient to support the beneficial effects of HIFT.

## Figures and Tables

**Figure 1 nutrients-17-01441-f001:**
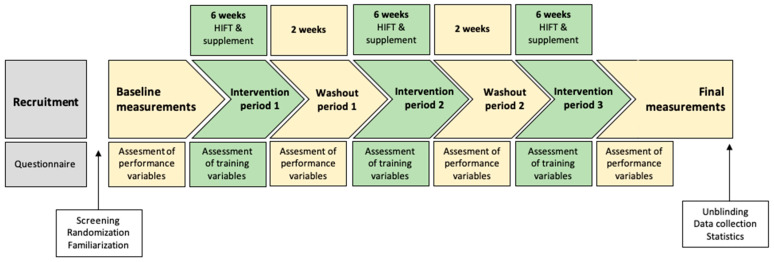
Study design.

**Table 1 nutrients-17-01441-t001:** Baseline characteristics of the participants (mean ± SD).

Variable	Males (*n* = 20)	Females (*n* = 10)
Age (y)	35.7 ± 8.5	30.2 ± 4.5
Body mass (kg)	84.8 ± 9.5	60.2 ± 6.9
Height (m)	1.81 ± 0.07	1.68 ± 0.05
Body mass index (kg/m^2^)	25.9 ± 2.5	21.4 ± 1.5
1 RM of shoulder press (kg)	64.8 ± 10.9	28.5 ± 5.7
Maximal force of shoulder muscles (Ν)	932 ± 295	396 ± 127
Maximal velocity of shoulder muscles (m/s)	3.24 ± 0.62	2.57 ± 0.58
Grip strength (Ν)	50.5 ± 7.4	28.8 ± 3.6
Strength endurance of core muscles (sit-ups/min)	37 ± 7	33 ± 4
Peak torque knee extensors 60°/s (Nm)	225 ± 47	122 ± 27
Peak torque knee flexors 60°/s (Nm)	133 ± 27	83 ± 16
Peak torque knee extensors 120°/s (Nm)	189 ± 39	103 ± 17
Peak torque knee flexors 120°/s (Nm)	119 ± 24	71 ± 12
Peak torque knee extensors 180°/s (Nm)	166 ± 31	86 ± 16
Peak torque knee flexors 180°/s (Nm)	103 ± 21	60 ± 9
Peak torque knee extensors 240°/s (Nm)	146 ± 27	73 ± 15
Peak torque knee flexors 240°/s (Nm)	90 ± 20	50 ± 10
Endurance ratio knee extensors (%)	74 ± 8	75 ± 10
Endurance ratio knee flexors (%)	68 ± 6	74 ± 8
VO_2_max (mL/kg/min)	46.3 ± 9.0	40.1 ± 4.4

**Table 2 nutrients-17-01441-t002:** Descriptive statistics of training variables during an HIFT session of the first and a HIFT session of the sixth week of each intervention period (mean ± SD for 13 males and 9 females).

	Egg White				Whey Protein				Placebo			
	Males		Females		Males		Females		Males		Females	
	1st Week	6th Week	1st Week	6th Week	1st Week	6th Week	1st Week	6th Week	1st Week	6th Week	1st Week	6th Week
Total exercise time (min:s)	40:01 ± 04:45	44:31 ± 06:53	44:17 ± 08:22	46:41 ± 11:57	41:50 ± 06:40	44:01 ± 08:43	45:41 ± 06:49	47:53 ± 07:50	38:21 ± 05:46	49:21 ± 08:17	41:59 ± 07:07	46:42 ± 09:16
Average HR (% HRmax)	76 ± 8	76 ± 7	75 ± 5	75 ± 8	74 ± 8	76 ± 6	76 ± 7	72 ± 7	72 ± 9	74 ± 8	73 ± 6	74 ± 6
Highest HR (% HRmax)	97 ± 9	96 ± 10	95 ± 7	95 ± 7	98 ± 11	93 ± 5	98 ± 8	98 ± 6	95 ± 16	100 ± 14	92 ± 4	97 ± 8
Time in HR zone 1 (min:s)	02:39 ± 03:18	03:47 ± 03:17	04:16 ± 02:38	05:46 ± 04:07	04:40 ± 05:03	04:52 ± 06:00	05:59 ± 06:29	07:33 ± 05:37	04:08 ± 04:15	06:34 ± 06:28	06:33 ± 04:07	05:23 ± 03:32
Time in HR zone 2 (min:s)	06:06 ± 03:36	08:23 ± 04:59	09:14 ± 06:00	10:19 ± 06:12	07:25 ± 03:48	07:36 ± 06:26	09:11 ± 04:37	10:12 ± 03:59	07:09 ± 04:23	09:48 ± 06:44	09:30 ± 05:41	08:34 ± 05:28
Time in HR zone 3 (min:s)	11:05 ± 04:04	11:25 ± 03:18	13:45 ± 04:32	11:56 ± 04:54	10:47 ± 04:57	12:13 ± 04:24	10:36 ± 02:49	11:34 ± 05:20	11:32 ± 07:56	12:34 ± 07:56	11:20 ± 06:04	10:02 ± 05:26
Time in HR zone 4 (min:s)	12:25 ± 05:48	12:38 ± 04:44	11:09 ± 04:52	11:55 ± 06:09	11:45 ± 05:57	13:24 ± 05:56	10:47 ± 04:13	10:47 ± 08:13	08:46 ± 06:19	12:21 ± 07:16	10:01 ± 05:45	13:04 ± 05:53
Time in HR zone 5 (min:s)	05:36 ± 07:01	06:04 ± 07:35	04:52 ± 05:06	05:34 ± 06:17	05:14 ± 06:29	05:06 ± 05:04	08:17 ± 07:33	04:24 ± 05:10	03:02 ± 04:36	05:14 ± 06:45	03:20 ± 04:46	04:32 ± 05:04
Energy expenditure (kcal)	513 ± 136	562 ± 121	365 ± 88	381 ± 115	514 ± 143	540 ± 101	385 ± 98	365 ± 117	437 ± 149	588 ± 165	326 ± 75	358 ± 107
Training load score	83 ± 30	89 ± 27	79 ± 16	82 ± 25	80 ± 33	86 ± 21	86 ± 29	75 ± 28	66 ± 30	88 ± 34	68 ± 19	77 ± 22
Cardio load	69 ± 29	74 ± 26	79 ± 16	83 ± 26	68 ± 29	72 ± 20	87 ± 28	77 ± 26	55 ± 25	75 ± 26	69 ± 19	78 ± 24

HR, heart rate.

**Table 3 nutrients-17-01441-t003:** Inferential statistics of training variables that produced significant outcomes by three-way ANOVA.

	Main Effect			Interaction			
	Supplement	Time	Sex	Supplement × Time	Supplement × Sex	Time × Sex	Supplement × Time × Sex
Total exercise time	0.828 0.009	<0.001 0.459	0.204 0.079	0.117 0.012	0.573 0.027	0.218 0.075	0.524 0.032
Time in HR zone 2 (min:s)	0.974 0.001	0.033 0.209	0.273 0.060	0.795 0.011	0.627 0.023	0.163 0.095	0.423 0.042
Time in HR zone 4 (min:s)	0.668 0.020	0.049 0.181	0.733 0.006	0.394 0.046	0.437 0.041	0.715 0.007	0.889 0.006
Energy expenditure	0.346 0.052	0.005 0.328	<0.001 0.444	0.084 0.116	0.903 0.005	0.025 0.228	0.504 0.034
Training load score	0.133 0.096	0.049 0.181	0.677 0.009	0.105 0.106	0.946 0.003	0.053 0.174	0.695 0.018
Cardio load	0.134 0.096	0.034 0.205	0.254 0.065	0.064 0.128	0.877 0.007	0.090 0.137	0.639 0.022

Within each cell, the top figure denotes the *p* value, and the bottom figure denotes the effect size (partial η^2^). Statistically significant findings (*p* ≤ 0.05) are highlighted.

**Table 4 nutrients-17-01441-t004:** Descriptive statistics of performance variables (mean ± SD for 20 males and 10 females).

	Egg White				Whey Protein				Placebo			
	Males		Females		Males		Females		Males		Females	
	Before	After	Before	After	Before	After	Before	After	Before	After	Before	After
1 RM of shoulder press (kg)	68.4 ± 11.1	68.8 ± 13.1	32.0 ± 5.5	32.9 ± 4.7	67.3 ± 11.1	69.0 ± 11.6	29.5 ± 6.2	32.2 ± 6.0	67.4 ± 13.2	70.3 ± 11.3	31.5 ± 5.7	31.5 ± 5.6
Maximal force of shoulder muscles (Ν)	894 ± 193	859 ± 187	396 ± 97	477 ± 125	913 ± 307	872 ± 158	409 ± 134	448 ± 95	858 ± 157	896 ± 182	499 ± 111	449 ± 118
Maximal velocity of shoulder muscles (m/s)	3.34 ± 0.81	3.47 ± 0.64	3.34 ± 1.34	2.54 ± 0.46	3.33 ± 0.57	3.54 ± 0.67	2.55 ± 0.74	2.56 ± 0.43	3.43 ± 0.57	3.36 ± 0.75	2.36 ± 0.45	3.14 ± 1.51
Grip strength (Ν)	51.0 ± 6.6	49.7 ± 7.1	28.4 ± 3.4	29.3 ± 2.8	50.2 ± 7.2	51.6 ± 6.3	29.1 ± 2.5	28.8 ± 2.3	49.6 ± 6.0	50.8 ± 6.0	28.9 ± 2.8	29.2 ± 2.5
Strength endurance of core muscles (sit-ups/min)	40 ± 6	42 ± 6	37 ± 4	40 ± 5	41 ± 7	42 ± 7	35 ± 4	37 ± 4	40 ± 7	42 ± 7	36 ± 5	40 ± 6
Peak torque knee extensors 60°/s (Nm)	214 ± 52	203 ± 43	114 ± 23	118 ± 18	208 ± 44	202 ± 46	124 ± 24	119 ± 21	207 ± 44	195 ±32	118 ± 18	113 ± 16
Peak torque knee flexors 60°/s (Nm)	136 ± 31	133 ± 30	80 ± 13	79 ± 14	132 ± 25	135 ± 28	82 ± 15	89 ± 37	133 ± 29	125 ± 24	79 ± 14	78 ± 12
Peak torque knee extensors 120°/s (Nm)	185 ± 44	180 ± 35	106 ± 18	106 ± 16	187 ± 38	176 ± 37	108 ± 19	107 ± 17	175 ± 28	178 ± 33	103 ± 13	96 ± 33
Peak torque knee flexors 120°/s (Nm)	120 ± 27	120 ± 25	74 ± 10	73 ± 11	119 ± 23	120 ± 22	72 ± 13	73 ± 11	119 ± 20	113 ± 19	71 ± 11	73 ± 10
Peak torque knee extensors 180°/s (Nm)	166 ± 34	162 ± 29	90 ± 19	91 ± 15	165 ± 30	163 ± 31	92 ± 16	92 ± 18	161 ± 24	162 ± 27	86 ± 14	91 ± 13
Peak torque knee flexors 180°/s (Nm)	108 ± 27	105± 21	63 ± 11	62 ± 10	104 ± 21	109 ± 26	61 ± 10	63 ± 10	104 ± 20	102 ± 17	62 ± 10	64 ± 10
Peak torque knee extensors 240°/s (Nm)	146 ± 31	144 ± 27	76 ± 17	76 ± 14	145 ± 27	145 ± 29	78 ± 16	79 ± 14	145 ± 26	144 ± 26	74 ± 11	79 ± 12
Peak torque knee flexors 240°/s (Nm)	96 ± 26	92 ± 21	53 ± 10	53 ± 11	90 ± 19	95 ± 26	51 ± 11	55± 10	93 ± 22	90 ± 16	53 ± 10	54 ± 9
Endurance ratio knee extensors (%)	73 ± 10	73 ± 9	74 ± 7	70 ± 9	74 ± 10	72 ± 9	72 ± 11	71 ± 7	73 ± 9	76 ± 11	72 ± 7	71 ± 8
Endurance ratio knee flexors (%)	68 ± 12	65 ± 10	68 ± 5	67 ± 6	65 ± 7	65 ± 10	71 ± 9	70 ± 6	65 ± 8	66 ± 15	71 ± 7	66 ± 6
VO_2_max (mL/kg/min)	47.2 ± 9.4	45.8 ± 7.6	40.5 ± 4.3	41.6 ± 4.5	46.9 ± 7.4	47.1 ± 8.4	40.9 ± 4.1	41.6 ± 4.4	46.5 ± 8.5	46.2 ± 8.3	42.5 ± 3.9	41.9 ± 3.5

1 RM, one-repetition maximum.

**Table 5 nutrients-17-01441-t005:** Inferential statistics of performance variables that produced significant outcomes by three-way ANOVA.

	Main Effect			Interaction			
	Supplement	Time	Sex	Supplement × Time	Supplement × Sex	Time × Sex	Supplement × Time × Sex
1 RM of shoulder press	0.302 0.041	<0.001 0.474	<0.001 0.767	0.472 0.026	0.564 0.018	0.430 0.022	0.270 0.046
Maximal force of shoulder muscles	0.571 0.017	0.449 0.021	<0.001 0.668	0.971 0.000	0.049 0.111	0.117 0.086	0.752 0.006
Maximal velocity of shoulder muscles	0.055 0.098	0.603 0.010	0.006 0.237	0.083 0.098	0.019 0.132	0.119 0.085	0.027 0.151
Grip strength	0.749 0.010	0.194 0.060	<0.001 0.806	0.589 0.019	0.695 0.013	0.823 0.002	0.133 0.070
Strength endurance of core muscles	0.348 0.037	<0.001 0.656	0.097 0.095	0.173 0.061	0.222 0.052	0.285 0.041	0.909 0.003
Peak torque knee extensors 60°/s	0.335 0.038	0.001 0.325	<0.001 0.604	0.759 0.010	0.409 0.031	0.027 0.163	0.659 0.015
Peak torque knee flexors 60°/s	0.075 0.088	0.664 0.007	<0.001 0.565	0.379 0.034	0.330 0.039	0.189 0.061	0.922 0.003
Peak torque knee extensors 120°/s	0.081 0.086	0.050 0.130	<0.001 0.635	0.864 0.005	0.900 0.004	0.683 0.006	0.454 0.028
Peak torque knee flexors 120°/s	0.322 0.040	0.435 0.022	<0.001 0.619	0.874 0.005	0.620 0.017	0.143 0.075	0.604 0.018
Peak torque knee extensors 180°/s	0.268 0.046	0.644 0.008	<0.001 0.688	0.635 0.016	0.934 0.002	0.108 0.090	0.944 0.002
Peak torque knee flexors 180°/s	0.576 0.019	0.468 0.019	<0.001 0.607	0.452 0.028	0.393 0.033	0.333 0.034	0.681 0.014
Peak torque knee extensors 240°/s	0.806 0.008	0.599 0.010	<0.001 0.679	0.754 0.010	0.797 0.008	0.146 0.074	0.907 0.003
Peak torque knee flexors 240°/s	0.922 0.003	0.774 0.003	<0.001 0.589	0.398 0.032	0.736 0.011	0.247 0.048	0.784 0.009
Endurance ratio knee extensors	0.892 0.003	0.202 0.058	0.544 0.013	0.456 0.028	0.541 0.020	0.043 0.138	0.470 0.027

1 RM, one-repetition maximum. Within each cell, the top figure denotes the *p* value, and the bottom figure denotes the effect size (partial η^2^). Statistically significant findings (*p* ≤ 0.05) are highlighted.

## Data Availability

Our data are provided free of charge and can be accessed via the following DOI: https://doi.org/10.26255/heal.ei79-mn9g (accessed on 24 April 2025).

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
