# Peer review of "Effects of Protein Supplementation During High-Intensity Functional Training on Physical Performance in Recreationally Trained Males and Females: A Randomized Controlled Trial"

_nutrients, 2025, doi:10.3390/nu17091441_

Round 1
Reviewer 1 Report
Comments and Suggestions for Authors
This cross-over trial examines the effects of additional high-quality protein (+0.6g/kg body weight; egg white and whey) on physical performance in recreationally trained adults. There were three treatments periods (four weeks each) in randomized order separated by two-week washout period. All participants were weight training prior to study enrollment. During the 4-week treatment periods, participants joined any available HIFT program (high intensity functional training-similar to CrossFit training) at their gyms for one-hour 3x weekly. Participants were encouraged to switch programs if they desired. During the washout periods, participants followed their typical (e.g., pre-study) weight training regimen. Given these protocol details, a cross-over study design seems unsuitable since concerns regarding carry-over and order effects are heightened. Indeed, the investigators noted carry-over effects for ‘duration of training session’ (increased from 42 to 47 minutes over the 16 weeks) and for HR (decreased from 141 to 135 bpm over the 16 weeks). The investigators state that these improvements indicated ‘positive adaptations’ to three 6-week training periods.
Specific concerns:
A possible correction to the carry-over concerns is to use separate baselines and control for treatment order. The investigators should also consider analyses of the first period only (e.g., parallel analyses) to avoid the confounding of the carry-over effect.
Diet plans were also changed every six weeks – hence each treatment phase was conducted under a different diet plan. To reduce confounding, diet could have been controlled for each 4-week treatment, and opened during the washouts. This is a study limitation that requires discussion. Please add a limitations section to the discussion.
Since the training followed by participants were not controlled, this also introduces a degree of variability that is a limitation to data interpretation. Please address.
Since participants continued weight training during the washout periods, training reversibility would likely not occur after only two weeks. Please address under limitations.
Line 60: it is unclear what Bagheri et al. results were – was this discussion accidently deleted?
Statistical analyses: were any outliers apparent? Were data normally distributed (as assumed for parametric analyses)?
Please proved a baseline descriptives table covering participant characteristics and strength markers.
Tables require information regarding sample size, statistical tests performed, etc. Is study phase controlled for in the analyses?
It is interesting that the additional protein did not have an impact on performance; however, the high variability linked to uncontrolled training regimens and the lack of true washout phases weaken confidence in data interpretation.
Author Response
Dear Reviewer,
Thank you for your constructive comments and for allowing us to improve our manuscript. Below, please find our point-by-point responses to your comments.
General comment: This cross-over trial examines the effects of additional high-quality protein (+0.6g/kg body weight; egg white and whey) on physical performance in recreationally trained adults. There were three treatments periods (four weeks each) in randomized order separated by two-week washout period. All participants were weight training prior to study enrollment. During the 4-week treatment periods, participants joined any available HIFT program (high intensity functional training-similar to CrossFit training) at their gyms for one-hour 3x weekly. Participants were encouraged to switch programs if they desired. During the washout periods, participants followed their typical (e.g., pre-study) weight training regimen. Given these protocol details, a cross-over study design seems unsuitable since concerns regarding carry-over and order effects are heightened. Indeed, the investigators noted carry-over effects for ‘duration of training session’ (increased from 42 to 47 minutes over the 16 weeks) and for HR (decreased from 141 to 135 bpm over the 16 weeks). The investigators state that these improvements indicated ‘positive adaptations’ to three 6-week training periods.
Response: Thank you for this comment. It is natural, expected, and desirable for a training program to result in positive adaptations like the increase in the duration of training sessions and decrease in HR. In anticipation of this, we applied RANDOMIZATION and COUNTERBALANCING of supplementation in our CROSS-OVER study, resulting in no differences in any baseline measurement. Otherwise, one would preclude any cross-over study with training interventions. Yet, the literature is rife with such studies in which the researchers exploit the great advantage of subjecting the same participants to all interventions (that is, controlling for inter-individual variability) and eliminate any carry-over bias by randomizing interventions. This approach enabled us to more accurately assess the potential effects of two forms of high-quality protein supplementation on adaptations to HIFT under conditions that closely reflect real-life practice. We believe that the chosen design was the most appropriate to address our research question in a setting that emphasizes ecological validity and practical relevance.
Specific comments:
Comment 1: A possible correction to the carry-over concerns is to use separate baselines and control for treatment order. The investigators should also consider analyses of the first period only (e.g., parallel analyses) to avoid the confounding of the carry-over effect.
Response: Our statistical analysis (3-way ANOVA) takes the separate baselines into account. Besides, as mentioned, this analysis detected no baseline differences between the three supplements, apparently because they were randomized. The randomized, counterbalanced, and cross-over design of the study controlled for any order effect. Analyzing the first period only would limit us to 10 participants per supplement and throw away two-thirds of our data, thus hugely weakening our study for no reason. Given the design integrity employed, we believe the current analysis remains robust and appropriate.
Comment 2: Diet plans were also changed every six weeks – hence each treatment phase was conducted under a different diet plan. To reduce confounding, diet could have been controlled for each 4-week treatment, and opened during the washouts. This is a study limitation that requires discussion. Please add a limitations section to the discussion.
Response 2: Please let us reiterate that the study was randomized and counterbalanced. Thus, with every dietary plan, one-third of the participants were taking egg white, one-third were taking whey protein, and one-third were taking placebo. As mentioned in the manuscript, dietary plans were changed for variety and to reflect real-world variability (especially considering seasonal changes in food availability). This is a regular practice followed by dietitians to prevent boredom and increase compliance. Please note that the dietary plans differed only in food sources, not macronutrient distribution (which was the same), and were always isoenergetic. We do not see how opening the diet during the washouts could reduce confounding, rather the opposite. Therefore, while we recognize the issue of dietary plans as a consideration, we do not view it as a limitation but rather as a deliberate methodological choice to increase the applicability of the study. We have added limitations to the one already described in the discussion (ll. 503–510).
Comment 3: Since the training followed by participants were not controlled, this also introduces a degree of variability that is a limitation to data interpretation. Please address.
Response: The training followed by the participants was controlled: As mentioned in ll. 167–169, certified instructors and/or research assistants, all carefully briefed by the investigators, supervised all sessions and checked the proper application of the exercise protocol, including the “work to momentary failure” principle. What was not controlled was the participants’ choice of the particular training schedule. This was a deliberate choice to reflect real-world conditions and enhance the ecological validity of the study. Standardizing training across participants could have introduced artificial constraints that do not align with typical practice. Therefore, while we recognize the variety in training contents (but within the limits of HIFT) as a factor, we do not consider it a limitation but rather an intentional aspect of the study design that strengthens the generalizability and applicability of the findings to real-life settings, as stated in ll. 555–557.
Comment 4: Since participants continued weight training during the washout periods, training reversibility would likely not occur after only two weeks. Please address under limitations.
Response: The washout periods were used to eliminate the effect of one supplement on the next, not of one training period on the next. We let the participants continue their regular training during the washout periods to keep them recreationally trained at the onset of the 2nd and 3rd periods, just like they were at the onset of the 1st period. Again, the study was randomized and counterbalanced, so it was important for all participants to be in the same state at the beginning of taking egg white, whey protein, or placebo.
Comment 5: Line 60: it is unclear what Bagheri et al. results were – was this discussion accidently deleted?
Response: As mentioned in the manuscript, Bagheri et al. studied egg-white supplementation without a non-protein control (the comparator was whole eggs of the same protein content as egg whites). Thus, they present no results concerning the effectiveness of egg white (since egg white was present in both interventions) and this is why we did not mention any results.
Comment 6: Statistical analyses: were any outliers apparent? Were data normally distributed (as assumed for parametric analyses)?
Response: When one examines the effects of more than one independent variable, as the three variables (supplementation, time, and sex) in our study, one has no alternative other than to use factorial ANOVA, since there is no non-parametric factorial test. In addition, it has been documented that factorial ANOVA is remarkably robust and resistant to violations of normality (Blanca et al., Psicothema 2017, 29, 552-557; Blanca et al., Psicothema 2023, 35, 21-29). Thus, although there were some scattered outliers and non-normally distributed data, we believe that our statistical conclusions are valid.
Comment 7: Please provide a baseline descriptives table covering participant characteristics and strength markers.
Response: Thank you for this suggestion. We have added the requested information as table 1.
Comment 8: Tables require information regarding sample size, statistical tests performed, etc. Is study phase controlled for in the analyses?
Response: Sample size is already stated in the tables’ titles. We have added the statistical tests performed to the tables’ titles. Yes, the study phase was controlled for in the analyses.
Final comment: It is interesting that the additional protein did not have an impact on performance; however, the high variability linked to uncontrolled training regimens and the lack of true washout phases weaken confidence in data interpretation.
Response: Based on what we wrote above, we do not believe that there was high variability linked to uncontrolled training regimens or a problem with the washout phases. Thank you again for your valuable comments, and we hope that we have addressed them to your satisfaction.
Reviewer 2 Report
Comments and Suggestions for Authors
Thank you for this fascinating study that filled a true gap in understanding of HIFT. As was stated in your abstract and introduction, even though HIFT is very popular there has been extremely limited research on how diet and dietary supplements impact HIFT. This study was well designed and reported with clarity. This reviewer has no recommended changes.
Author Response
Thank you very much for your kind comments and for approving our study.
Reviewer 3 Report
Comments and Suggestions for Authors
While a study itself is of some merit, especially because it is interventional and some important aspects had been attempted to to put under control, the paper lacks rigor and need substantial revisions.
Why do you call the study triple blinded if the participants were not blinded as you say? I suggest to use more proper term here ("researcher blinded" or something of that category - please consult the relevant literature for categorization of such type of studies).
Were the participants really aware they were ingesting placebo (carbohydrates) or (whey) proteins? Please extricate, and also describe the formulations the three supplements they were consuming (whey powder dissolved in water? what about maltodextrin? and especially egg white -- you say it was liquid, but how processed; and consumed alone or with another carrier ? etc.). What was the timing between supplementation and meals, and how it was judged on and controlled for ?
Were participants recreationally trained or "trained"? Please describe them properly and then use the term consistently.
Where the tests had been conducted? Please provide description for the organization of the study properly. Also related to that - where thigh muscle function has been tested (I guess not in the gym but in the lab as well as VO2max test). What was the order of the tests, the warm ups, the distribution of the tests between days etc.
The Results section way too bulky and too many (secondary) indices are presented o it is hard to read. Please provide the major parameters/findings in the graphs and avoid large tables. Tables 2 and 4 are especially suggested to be put into Suppl. materials (what they give of importance in fact? there are statistical significances indicated but not clear which group responded stringer), and Tables 1 and 3 needs to be condensed if decided to be left in the main text. The tables need p-values where they were <0.05 at least.
Author Response
Dear Reviewer,
Thank you for your constructive comments and for allowing us to improve our manuscript. Below, please find our point-by-point responses to your comments.
Comment 1: Why do you call the study triple blinded if the participants were not blinded as you say? I suggest to use more proper term here ("researcher blinded" or something of that category - please consult the relevant literature for categorization of such type of studies).
Response: Thank you for raising this important point. We called the study triple-blinded because it was blind regarding the intervention providers (HIFT instructors and supplement distributors), outcome assessors, and investigators, as stated in ll. 90–91 and according to the literature. However, following your suggestion, we have replaced the term with “researcher-blinded” for clarity.
Comment 2: Were the participants really aware they were ingesting placebo (carbohydrates) or (whey) proteins? Please extricate, and also describe the formulations the three supplements they were consuming (whey powder dissolved in water? what about maltodextrin? and especially egg white -- you say it was liquid, but how processed; and consumed alone or with another carrier ? etc.). What was the timing between supplementation and meals, and how it was judged on and controlled for?
Response: Thank you for this comment. The participants were not aware of whether they were ingesting a placebo or whey protein powder. We have added clarification of this in ll. 93 and the formulations in ll. 142–145. As mentioned in l. 142, the supplements were taken in two equal daily doses, approximately 12 h apart. The timing between supplementation and meals was not controlled since there is no information in the literature that it plays a role in the long run.
Comment 3: Were participants recreationally trained or "trained"? Please describe them properly and then use the term consistently.
Response: Thank you for this comment. The participants were recreationally trained. We now use this term consistently throughout the manuscript.
Comment 4: Where the tests had been conducted? Please provide description for the organization of the study properly. Also related to that - where thigh muscle function has been tested (I guess not in the gym but in the lab as well as VO2max test). What was the order of the tests, the warm ups, the distribution of the tests between days etc.
Response: Thank you for this comment. As stated in ll. 179–180, all tests (including thigh muscle function and VO2max test) were carried out at the Laboratory of Evaluation of Human Biological Performance. We have added detailed information on how they were organized (order of tests, warm-ups, and distribution between days) in ll. 185-192.
Comment 5: The Results section way too bulky and too many (secondary) indices are presented o it is hard to read. Please provide the major parameters/findings in the graphs and avoid large tables. Tables 2 and 4 are especially suggested to be put into Suppl. materials (what they give of importance in fact? there are statistical significances indicated but not clear which group responded stringer), and Tables 1 and 3 needs to be condensed if decided to be left in the main text. The tables need p-values where they were <0.05 at least.
Response: We appreciate your concern about clarity and readability. Therefore, we have removed absolute heart rate values and percentages of time expended in each time zone from tables 2 and 3 (formerly tables 1 and 2, renumbered after the insertion of a new table 1 with baseline characteristics at the request of the reviewer 1). Additionally, we have removed the variables that did not exhibit statistically significant outcomes from the tables with inferential statistics. These tables provide critical context for understanding the patterns of response across all independent variables (including the p values that you request) in one place. Statistically significant outcomes are then explained in the text in terms of which group responded stronger etc. The tables also help illustrate interactions that are not easily captured in graphical form. Additionally, the tables contain effect sizes, which are important and can prove valuable to a meticulous reader or a future review or meta-analysis.
Thank you again for your valuable comments, and we hope that we have addressed them to your satisfaction. In response to your comment that English could be improved, we made linguistic changes with the aid of a native English speaker.
Round 2
Reviewer 3 Report
Comments and Suggestions for Authors
Overall, you have addressed my comments, even if somewhat formally for some points.,